# Incidence and Patterns of Digestive Organ Cancer in Georgia: Insights from a Population-Based Registry Study in 2021

**DOI:** 10.3390/jpm13071121

**Published:** 2023-07-10

**Authors:** Miranda Nonikashvili, Maia Kereselidze, Otar Toidze, Tina Beruchashvili

**Affiliations:** 1Department of Public Health and Healthcare Management, School of Health Sciences, The University of Georgia, Tbilisi 0175, Georgia; m.kereselidze@ug.edu.ge (M.K.); o.toidze@ug.edu.ge (O.T.); t.beruchashvili@ug.edu.ge (T.B.); 2Department of Medical Statistics, National Center for Disease Control and Public Health, Tbilisi 0198, Georgia

**Keywords:** digestive organ cancer, cancer registry, public health

## Abstract

**Simple Summary:**

A population-based registry study was conducted to investigate the incidence of digestive organ cancer in Georgia. The study used data from the Georgian Population-based Cancer Registry (GPCR) and included all cases of digestive organ cancer diagnosed in Georgia in 2021. The results showed that the overall incidence of digestive organ cancer in Georgia was similar to the global average, but there were some differences in the specific types of digestive organ cancer that were most common in the country. The study also found that the incidence of digestive organ cancer varied by region in Georgia, with some areas having higher rates of this disease than others. The results of this study provide important insights into the incidence of digestive organ cancer in Georgia and highlight the need for further research to better understand the factors that contribute to this disease.

**Abstract:**

Digestive organ cancer, also known as gastrointestinal (GI) cancer, refers to cancer that occurs in the digestive tract. In this population-based registry study, we aimed to investigate the incidence of GI in Georgia and to identify any patterns in the occurrence of different types of this disease. The study included all cases of GI cancer that were diagnosed in Georgia in 2021. We analyzed 1635 patients’ data to determine the overall and age-standardized incidence of GI cancer in the country. The analyses were performed for esophagus, stomach, colon, rectum, anus, liver and intrahepatic bile ducts, gallbladder, and pancreas separately. The descriptive statistics used in the study—specifically counts, means, proportions, and rates—were calculated using the statistical software STATA version 17.0. (StataCorp, College Station, TX, USA). The results of the study showed that the incidence of digestive organ cancer in Georgia was similar to the global average. However, there were some notable differences in the specific types of GI cancer that were most common in the country. Overall, this study provides important insights into the incidence of digestive organ cancer in Georgia and highlights the need for further research to better understand the factors that contribute to this disease.

## 1. Introduction

Cancer of the digestive organs, also known as gastrointestinal (GI) cancer, refers to cancer that occurs in the digestive tract, which includes the esophagus, stomach, small intestine, large intestine (colon and rectum), liver, pancreas, and gallbladder [1]. GI cancer can be a serious and life-threatening disease, but it can often be treated effectively if detected early. Digestive tract cancers are among the most common types of tumors, affecting over 5.1 million people globally and causing 3.6 million deaths in 2020 [2]. They are especially prevalent in Central and Eastern Europe [2,3]. Reducing the negative health consequences of these cancers will necessitate the collection of data that will lead to feasible prevention measures.

Georgia is an upper middle-income country located between Europe and Asia, with a population of 3.7 million people [4]. It has a GDP per capita of USD 5023, which is slightly above the lower threshold of upper-middle-income countries [5,6]. The country of Georgia acquired independence following the dissolution of the Soviet Union in 1991 [7]. Georgian society has changed dramatically in recent decades. The population’s living and dietary habits have changed as society and the economy have progressed, and this, combined with an aging population, has resulted in digestive organ cancer being a significant social and economic burden.

Cancer is a significant disease burden in Georgia among noncommunicable diseases, and it is often diagnosed at a late stage. The Georgian Cancer Registry was established in 2015 as a population-based national registry. Therefore, reports are prepared from the date of establishment; however, few studies have been conducted using registry data. Moreover, sufficient knowledge is not yet available, and the figures provided are age-standardized using Georgia’s population, preventing comparison with international data. In Georgia, there were 7873 cancer cases in 2019. Lung (15%) and breast (10%) cancers were the leading causes, followed by cancers of the trachea, bronchus, digestive system (excluding stomach and esophagus), lymphoid, hematopoietic and related tissue, and colorectal cancer [8].

It is important to study and understand the prevalence of digestive organ cancer in the country of Georgia because it can help to inform the development and implementation of public health policies and initiatives aimed at reducing the burden of this disease. Understanding the specific types of digestive organ cancer that are most common in Georgia, as well as the risk factors that contribute to their development, can help to target prevention efforts and allocate resources effectively. Additionally, understanding the patterns of digestive organ cancer in Georgia can help to identify potential areas for further research and provide insight into possible strategies for early detection and treatment of this disease.

By addressing these research questions, the study aims to provide valuable insights into the burden of digestive organ cancer in Georgia. The findings have clinical relevance as they can contribute to the understanding of the disease, guide healthcare planning and resource allocation, and aid in the development of targeted prevention and treatment strategies. Moreover, the study may help to identify research gaps and inform public health policies aimed at reducing the incidence and impact of digestive organ cancer in the population of Georgia.

To the best of our knowledge, no studies have examined digestive organ cancer patterns in the country of Georgia. In this population-based registry study, we aimed to investigate the incidence of digestive organ cancer in Georgia with a focus on regional disparities. By understanding any differences in the prevalence of this disease between different regions of the country, we can identify potential areas for further research and potentially target prevention efforts in order to reduce the burden of digestive organ cancer in Georgia. We hope to assist in identifying priority areas for research and development in this field.

## 2. Materials and Methods

### 2.1. Study Population

Data on cancer incidence were gathered from Georgia’s population-based cancer registry. The study population for this study consisted of all individuals who were diagnosed with digestive organ cancer in Georgia in 2021. The data for this study were obtained from the Georgian Population-based Cancer Registry (GPCR), which is a comprehensive database of cancer cases in the country. The GPCR includes information on all newly diagnosed cancer cases in Georgia, including information on the type of cancer, the stage of the cancer at diagnosis, and the patient’s demographic characteristics (such as age, sex, race/ethnicity, etc.). These data were used to identify all cases of digestive organ cancer that were diagnosed in Georgia during the study period.

### 2.2. Site Classification

In this population-based registry study, the site of the digestive organ cancer was classified according to the International Classification of Diseases for Oncology, Third Edition (ICD-O-3) [9]. The following codes were used: esophageal (C15); stomach (C16); colorectal and anal (C18, C19–C20, C21); liver and intrahepatic bile duct (C22); gallbladder (C23); and pancreatic cancer (C25).

### 2.3. Statistical Analysis

In this population-based registry study, statistical analysis was used to investigate the incidence of digestive organ cancer in Georgia and to identify associated characteristics in the occurrence of different types of this disease. The following statistical methods were used in the analysis. Descriptive statistics were used to summarize the characteristics of the study population, including the age and gender distribution of the individuals with digestive organ cancer. For example, descriptive statistics were used to calculate means, standard deviations, and 25th and 75th percentiles for continuous variables and percentages for categorical and binary variables. Incidence rates were calculated to determine the overall incidence of digestive organ cancer in Georgia. Incidence rates were also calculated for specific types of digestive organ cancer (such as stomach cancer, colon cancer, etc.). Additionally, the European Standard Population (ESP) and the World Standard Population (WSP) were used as a reference for calculating age-standardized rates [10,11]. In the context of the incidence of digestive organ cancer in Georgia, the use of the ESP and WSP allows for the comparison of the incidence of this disease in Georgia to the average incidence in European countries and the world population, respectively. The chi-square test was used to test for statistical differences in the incidence of digestive organ cancer between different subgroups of the population (such as men vs. women or different age groups). The statistical software Stata 17 (StataCorp. 2021. Stata Statistical Software: Release 17. College Station, TX, USA: StataCorp LLC) was used to analyze the data.

### 2.4. Ethical Approval

The study protocol was revised and approved by the NCDC&PH Institutional Review Board (IRB N 2022-009/IRB0000215 21.02.2022).

## 3. Results

In 2021, the average age at diagnosis of the digestive organ cancer patients was 65.1 years (IQR: 59–73 years). The age-standardized rate for colon cancer was 5.3 per 100,000 total population, and rectal cancer and stomach cancer rates were 4.3 per 100,000 and 4.2 per 100,000, respectively.

The majority of cancers were registered among males (57.4%). The most commonly reported age group was the 50–69 years category (55.1%). Of the registered cases, 39.3% were from Western Georgia and 30.1% were from the capital city of Georgia, Tbilisi. However, three out of five (62.2%) registered cases were from an urban area (Figure 1).

Colorectal (C18–C21) and stomach (C16) cancer were most commonly registered cancer among both sexes. Liver cancer was most commonly diagnosed cancer among males (76.9%). However, gallbladder cancer cases were significantly higher among females (70.5%). Liver and intrahepatic bile duct cancer was 3.3 times higher (66.8%) among patients in the 50–69 years age group. About half of the esophagus cancer cases were registered in Eastern Georgia (48.0%); however, 45% of gallbladder and pancreatic cancers were diagnosed among the Western Georgian population. There were no settlement type differences in the frequency of stomach cancer cases, but for the majority of the rest of the cancer sites the patients’ settlement type was urban; for example, colon cancer cases were two times higher (66.0%) in urban areas than in rural areas (Figure 1, Table 1 and Table 2).

The sample population consisted of 57.43% (n = 939) males and 42.57% (n = 696) females. The age of the study participants ranged from 2 to 93 years, with a mean age of 65.16 ± 11.49 years. Among the participants, 55.11% fell within the age group of 50–69 years.

The results show that esophageal cancer primarily affects individuals in the age group 50–69 years. Stomach cancer exhibits a relatively even distribution across age groups with a total of 337 cases. Colon cancer shows the highest number of cases (423) among individuals aged ≤49 years. Rectal cancer is most prevalent among individuals aged ≥70 years with a total of 358 cases. Anal cancer predominantly occurs in the age group 50–69 years with 15 reported cases; the median age of patients is 63 years. Liver and intrahepatic bile duct cancer primarily affects individuals aged 50–69 years with a total of 226 cases, and the median age of patients is 60 years. There are 34 gallbladder cancer cases with the majority occurring in individuals aged ≥70 years, and the median age of patients is 63 years. Pancreatic cancer shows a higher prevalence among individuals aged ≤49 years with 190 cases; the median age of patients is 61 years (Table 2).

Table 3 provides a summary of histological groups and their distribution across different anatomical locations. The results show that carcinoma is the most prevalent histological group in all locations, ranging from 46.5% in the pancreas to 93.3% in the anus. Squamous cell carcinoma is prominent in the esophagus, anus, and gallbladder, while adenocarcinoma is predominant in the stomach, colon, rectum, and pancreas. Unspecified malignant neoplasms are relatively common in the liver and intrahepatic bile ducts and pancreas.

The geographical distribution of gastrointestinal (GI) incidence across different regions in Georgia shows varying rates. Ajaria has the highest GI incidence rate of 53.7. Tbilisi, the capital city, has a moderate GI incidence rate of 39.5, while Guria, Kakheti, and Samegrelo-Zemo Svaneti have rates ranging from 40 to 41. Imereti and Shida Kartli have slightly higher rates of 43.8 and 42.8, respectively. Mtskheta-Mtianeti has the lowest GI incidence rate of 33.3 (Figure 2).

The chi-square test revealed a statistically significant difference in the incidence of GI cancer between the different sexes, ages, regions, and settlement types (*p* < 0.05) (Table 4).

## 4. Discussion

The results of this population-based registry study showed that the incidence of digestive organ cancer in Georgia was similar to the global average. However, there were some notable differences in the specific types of digestive organ cancer that were most common in the country. For example, Georgia had a higher-than-average incidence of stomach cancer, which was the third most common type of digestive organ cancer in the country. *Helicobacter pylori* was the leading cause of stomach cancer, accounting for nearly 90% of new cases of noncardia gastric cancer [12,13]. There was a dietary component, with salt-preserved foods and a lack of fruits increasing risk, and both alcohol consumption and active tobacco smoking are established risk factors [14]. In contrast, the incidence of colorectal cancer, which is the third most common type of digestive organ cancer globally, was lower in Georgia compared to the global average. These differences may be related to differences in the late diagnosis issue, diet, lifestyle, and other risk factors between Georgia and other countries. Moreover, low screening uptake and late detection could be one of the main reasons for this finding [15,16]. Further research is needed to better understand the factors that contribute to the development of digestive organ cancer in Georgia.

The results also showed that the incidence of digestive organ cancer varied by region and settlement type in Georgia, with some areas having higher rates of this disease than others. The spatial analysis revealed that the highest rates of digestive organ cancer were concentrated in urban areas. The differences in the incidence of digestive organ cancer in Georgia compared to other countries and in urban areas compared to rural areas can be attributed to a combination of factors such as variations in demographic and lifestyle factors, differences in environmental exposures, disparities in healthcare access and awareness, and variations in data collection and reporting practices [17,18]. The study revealed variations in the incidence rates of digestive organ cancers across different regions of Georgia. Tbilisi, Ajaria, and Imereti were identified as regions with relatively higher incidence rates, while Mtskheta-Mtianeti and Samtskhe-Javakheti had lower rates. These regional disparities in incidence rates may be attributed to a combination of factors, including differences in risk factors, socioeconomic status, and access to healthcare services. Understanding these regional variations can help tailor cancer control strategies and allocate resources more effectively [19,20,21,22]. Overall, the distribution reveals regional differences in GI incidence, highlighting the need for further investigation into potential factors contributing to these variations. Furthermore, the findings provide valuable insights for public health planning and policy development. The identification of regions with higher incidence rates and specific cancer types enables the prioritization of resources for early detection, screening, and treatment interventions. Additionally, these results can guide public health campaigns and educational initiatives aimed at promoting healthy lifestyles and raising awareness about the risk factors associated with digestive organ cancers.

Liver cancer was the fourth most commonly diagnosed cancer in Georgia. Liver cancer, specifically hepatocellular carcinoma (HCC), is a major public health concern worldwide [23]. In Georgia, as in many other countries, chronic infection with hepatitis C virus (HCV) and hepatitis B virus (HBV) are major risk factors for the development of HCC. In the country of Georgia, efforts to eliminate hepatitis C virus (HCV) infection have been underway for several years. The Georgian government has implemented a number of strategies to increase access to HCV testing and treatment, including establishing a network of HCV treatment centers and negotiating lower prices for HCV antiviral drugs. This program has helped to improve record keeping of liver cancer in recent years [24].

In terms of demographic characteristics, the study found that the incidence of digestive organ cancers was higher in men than in women except for gallbladder cancer. Gallbladder cancer, unlike other types of digestive organ cancer, was more common in women than in men.

The higher prevalence of pancreatic cancer in the age group of ≤49 years suggests potential contributions from genetic or hereditary factors, as well as socioeconomic and lifestyle factors. Further investigations are needed to explore the role of genetic mutations and hereditary syndromes in younger individuals with pancreatic cancer [25]. Additionally, the influence of socioeconomic factors and lifestyle choices should be examined to understand their impact on disease development. Addressing diagnostic challenges, such as improving early detection methods, will also be crucial for accurately assessing the prevalence of pancreatic cancer in this age group [26]. Further research is warranted to gain deeper insights into these factors and improve prevention and diagnostic strategies.

The trends and patterns of specific cancer subtypes vary based on their etiology and causal pathways. For instance, there are contrasting trends between esophageal adenocarcinoma (AC) and esophageal squamous cell carcinoma (SCC), with AC increasing in many high-income countries while SCC is declining. Similarly, hepatocellular carcinoma (HCC) is increasing in high-income countries but decreasing in most low-risk countries, while intrahepatic cholangiocarcinoma (ICC) rates appear to be rising in various countries regardless of income level [27].

The incidence of digestive organ cancer also increased with age, with the highest rates occurring in individuals over the age of 65 years in Georgia. The study’s findings can have a substantial impact on clinical practice. By examining the incidence rates and patterns of digestive organ cancer, healthcare providers can better understand the disease burden in Georgia and tailor their diagnostic, treatment, and prevention strategies accordingly. The identification of specific types of digestive organ cancer that are most prevalent in the population can guide physicians in early detection and timely interventions, leading to improved patient outcomes. Furthermore, the study’s results can inform public health initiatives and policies. By understanding the epidemiological landscape of digestive organ cancer in Georgia, policymakers can allocate resources more effectively to areas with higher incidence rates or specific populations at greater risk. The findings can also assist in implementing targeted public health campaigns aimed at raising awareness, promoting early screening, and adopting preventive measures against digestive organ cancer.

### Strength and Limitations

All analyses were based on routinely collected data. The study utilized a population-based registry, which is considered a robust data source for assessing cancer incidence. The paper focused on the incidence of digestive organ cancer, covering multiple sites such as the esophagus, stomach, colon, rectum, anus, liver, gallbladder, and pancreas. This comprehensive coverage provides a holistic understanding of the burden of digestive organ cancers in the study population. The study provided a detailed analysis of cancer incidence by age groups (≤49, 50–69, and ≥70 years) and sex.

However, our study also has several limitations. First of all, owing to the retrospective nature of our analyses we were unable to collect some important information. For example, we could not analyze metastatic burden and describe metastatic sites. The other major limitations of this study are the unmeasured established risk factors. The study focuses on the incidence and patterns of digestive organ cancer in 2021, which limits the understanding of long-term trends or changes over time. The population-based cancer registry data in Georgia may not capture detailed clinical information about individual cases, such as treatment modalities, or patient outcomes. The study may not have accounted for all potential confounding variables that could influence the observed associations between risk factors and digestive organ cancer incidence.

Overall, the results of this population-based registry study provide an updated overview of the incidence of digestive organ cancer in Georgia. The findings highlight the need for continued monitoring of this disease and the development of targeted interventions to reduce the burden of GI cancer in Georgia. Further research is necessary to better understand the factors contributing to the incidence of digestive organ cancer and to inform the design of effective public health strategies.

## 5. Conclusions

In conclusion, this population-based registry study found that the incidence of digestive organ cancer in Georgia is similar to the global average. However, there are some notable differences in the specific types of digestive organ cancer that are most common in the country, with higher-than-average rates of stomach cancer and lower-than-average rates of colorectal cancer. The average age at diagnosis was 65.1 years, with the most commonly reported age group being 50–69 years. The majority of cancers were registered among males, and the most prevalent cancer sites were colorectal and stomach cancer for both sexes. The geographic distribution showed a higher concentration of cases in Western Georgia and the capital city of Tbilisi, with a majority of cases occurring in urban areas. The histological analysis revealed that carcinoma was the most common histological group across all locations, followed by squamous cell carcinoma and adenocarcinoma. The study findings underscore the importance of targeted strategies for early detection, prevention, and public health interventions to reduce the burden of digestive organ cancer in Georgia.

Further research is needed to understand the factors that contribute to these patterns and to develop effective strategies for the prevention, early detection, and treatment of digestive organ cancer in Georgia. It is important for public health professionals and policymakers in the country to be aware of the incidence and patterns of this disease in order to take appropriate action to address it.

## Figures and Tables

**Figure 1 jpm-13-01121-f001:**
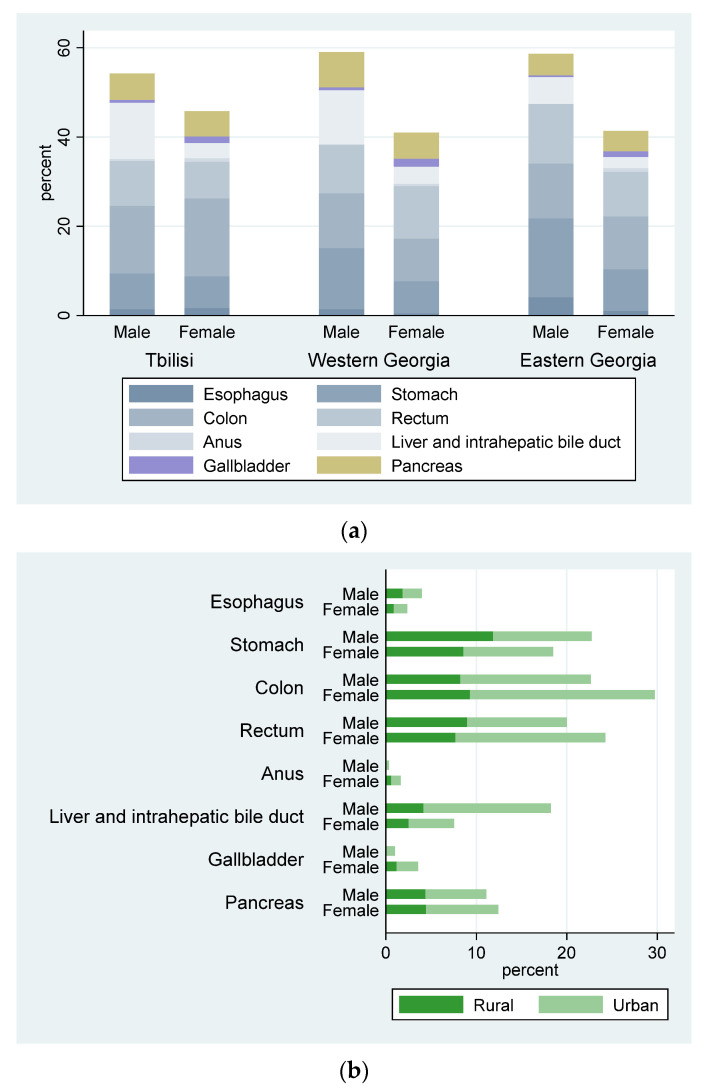
(**a**) Digestive organ cancers by sex and region, Georgia, 2021. (**b**) Digestive organ cancer by sex and settlement type, Georgia, 2021.

**Figure 2 jpm-13-01121-f002:**
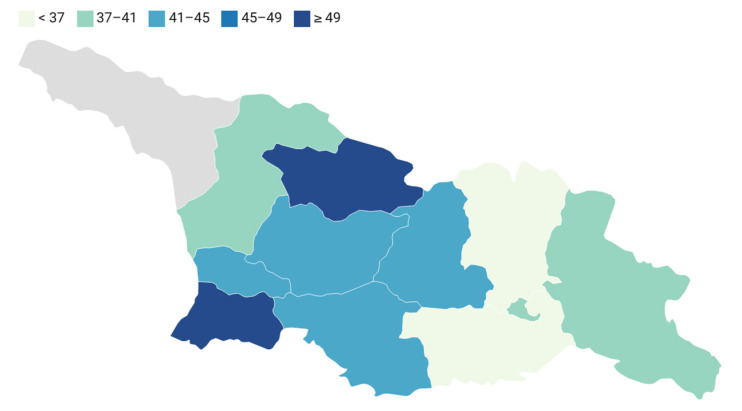
Incidence rate of digestive organ cancers in Georgia by regions in 2021 per 100,000 population.

**Table 1 jpm-13-01121-t001:** Crude and Age-standardized incidence rates (ESR and WSR) by digestive organ cancer site, Georgia, 2021.

Cancer Site (ICD-10)	CR ^1^	ESR ^2^	WSR ^3^
Esophagus (C15)	1.4	1.6	0.6
Stomach (C16)	9.0	10.4	4.2
Colon (C18)	11.4	13.2	5.3
Rectum (C19–20)	9.6	11.2	4.3
Anus (C21)	0.4	0.4	0.1
Liver and intrahepatic bile ducts (C22)	6.0	6.7	3.1
Gallbladder (C23)	0.9	1.0	0.3
Pancreas (C25)	5.1	5.7	2.5

^1^ CR = Crude (all ages) Rate (N/100,000 person-years); ^2^ ESR (1976) = Age-Standardized Rate, using the European Standard Population (N/100,000 person-years); ^3^ WSR = Age-Standardized Rate, using the World Standard Population (N/100,000 person-years).

**Table 2 jpm-13-01121-t002:** Digestive organ cancer cases by site and subsite in Georgia, 2021.

Cancer Site and Subsite	ICD-10 Code	Overall	≤49	50–69	≥70	Median (IQR)	Male	Female
Esophagus	C15	52	5 (3.62%)	28 (3.11%)	19 (3.19%)	63 (60–66)	36 (3.83%)	16 (2.30%)
Stomach	C16	337	24 (17.39%)	187 (20.75%)	126 (21.14%)	62 (58.5–66)	211 (22.47%)	126 (18.10%)
Colon	C18	423	39 (28.26%)	226 (25.08%)	158 (26.51%)	62 (59–66)	214 (22.79%)	209 (30.03%)
Rectum	C19, C20	358	26 (18.84%)	175 (19.42%)	157 (26.34%)	62 (59–66)	186 (19.81%)	172 (24.71%)
Anus	C21	15	1 (0.72%)	10 (1.11%)	4 (0.67%)	63 (60–68)	3 (0.32%)	12 (1.72%)
Liver and intrahepatic bile ducts	C22	226	18 (13.04%)	151 (16.76%)	57 (9.56%)	60 (57–64)	174 (18.53%)	52 (7.47%)
Gallbladder	C23	34	1 (0.72%)	20 (2.22%)	13 (2.18%)	63 (59–67)	10 (1.06%)	24 (3.45%)
Pancreas	C25	190	24 (17.39%)	104 (11.54%)	62 (10.40%)	61 (58–67)	105 (11.18%)	85 (12.21%)

Note: The percentages are calculated based on the total number of cases for each cancer site and subsite.

**Table 3 jpm-13-01121-t003:** Distribution of histological groups across anatomical locations of digestive organ cancer cases in Georgia, 2021.

Histological Groups	Esophagus (C15) n (%)	Stomach (C16) n (%)	Colon (C18) n (%)	Rectum (C19–C20) n (%)	Anus (C21) n (%)	Liver and Intrahepatic Bile Ducts (C22) n (%)	Gallbladder (C23) n (%)	Pancreas (C25) n (%)
Carcinoma	43 (82.7%)	271 (80.4%)	377 (89.9%)	316 (88.2%)	14 (93.3%)	83 (36.7%)	24 (70.6%)	88 (46.5%)
Squamous cell carcinoma	35 (67.3%)	3 (0.9%)	3 (0.7%)	13 (3.6%)	9 (60.0%)	23 (10.2%)	18 (52.9%)	12 (6.3%)
Adenocarcinoma	6 (11.5%)	252 (74.7%)	372 (88.8%)	297 (82.9%)	5 (33.3%)	60 (26.5%)	6 (17.6%)	76 (40.2%)
Other specified carcinoma	-	-	-	-	-	-	-	-
Unspecified carcinoma	2 (3.8%)	15 (4.4%)	2 (0.4%)	6 (1.6%)	-	-	-	-
Hepatoblastoma	-	-	-	-	-	2 (0.9%)	-	-
Sarcoma	2 (3.8%)	10 (2.9%)	6 (1.4%)	2 (0.5%)	-	3 (1.3%)	-	-
Other specified malignant neoplasm	-	-	-	-	-	-	-	-
Unspecified malignant neoplasm	7 (13.5%)	50 (14.8%)	36 (8.6%)	38 (10.6%)	1 (6.6%)	138 (61.1%)	10 (29.4%)	101 (53.4%)

**Table 4 jpm-13-01121-t004:** Characteristics of digestive organs cancer patients in Georgia, 2021.

Variable	Esophagus (C15) n (%)	Stomach (C16) n (%)	Colon (C18) n (%)	Rectum (C19–C20) n (%)	Anus (C21) n (%)	Liver and Intrahepatic Bile Ducts (C22) n (%)	Gallbladder (C23) n (%)	Pancreas (C25) n (%)	*p* Value *
Sex									<0.000
Male	36 (69.2)	211 (62.6)	214 (50.5)	186 (51.9)	<5	174 (76.9)	10 (29.4)	105 (55.2)	
Female	16 (30.7)	126 (37.3)	209 (49.4)	172 (48.0)	12 (80.0)	52 (23.0)	24 (70.5)	85 (44.7)	
Age									<0.007
Birth to 49	5 (9.6)	24 (7.1)	39 (9.2)	26 (7.2)	<5	18 (7.9)	<5	24 (12.6)	
50–69	28 (53.8)	187 (55.4)	226 (53.4)	175 (48.8)	10 (66.6)	151 (66.8)	20 (58.8)	104 (54.7)	
≥70	19 (36.5)	126 (37.3)	158 (37.3)	157 (43.8)	<5	57 (25.2)	13 (38.2)	62 (32.6)	
Region									<0.000
Tbilisi (Capital)	15 (28.8)	72 (21.6)	155 (38.1)	87 (25.6)	6 (42.8)	76 (35.1)	10 (30.3)	55 (29.7)	
Western Georgia	12 (23.0)	130 (39.1)	135 (33.2)	140 (41.3)	<5	99 (45.8)	15 (45.4)	85 (45.9)	
Eastern Georgia	25 (48.0)	130 (39.1)	116 (28.5)	112 (33.0)	<5	41 (18.9)	8 (24.2)	45 (24.3)	
Settlement Type									<0.000
Rural	23 (44.2)	166 (50.0)	138 (33.9)	134 (38.7)	<5	55 (25.3)	9 (27.2)	70 (37.8)	
Urban	29 (55.7)	166 (50.0)	269 (66.0)	212 (61.2)	10 (71.4)	162 (74.6)	24 (72.7)	155 (62.1)	

* *p*-values for difference: Chi-square tests between digestive organ cancers.

## Data Availability

The data presented in this study are available on request from the corresponding author.

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
