# Peer review of "Incidence and Patterns of Digestive Organ Cancer in Georgia: Insights from a Population-Based Registry Study in 2021"

_jpm, 2023, doi:10.3390/jpm13071121_

Round 1

Reviewer 1 Report

Purpose of the study and its clinical relevance with the patient population can be improved.

Tumors just based location without histological subtypes gives little benefit, can histological subtypes be added ?

No reference given for type of hepatitis infection in Georgia.

Discussion and Conclusion can be improved by discussing reasons for the results.

Reviewer 2 Report

Thank you for the privilege of reviewing your work. This manuscript is well written. I think the manuscript needs a little more consideration.

1.      The authors must add the limitation.

2.      Why did pancreatic cancer show a higher prevalence aged 49?

3.      Please describe the reason for the differences in incidence in Georgia compared to other countries and in urban areas compared to in rural areas in more detail.

4.      What were the strengths of this paper?

English is well written

Round 2

Reviewer 2 Report

The authors adequately revised the manuscript. I have no further comments.